# Research on Distributed Energy Consensus Mechanism Based on Blockchain in Virtual Power Plant

**DOI:** 10.3390/s22051783

**Published:** 2022-02-24

**Authors:** Dewen Wang, Zhao Wang, Xin Lian

**Affiliations:** 1School of Control and Computer Engineering, North China Electric Power University, Baoding 071003, China; wdewen@gmail.com (D.W.); lianxin0302@163.com (X.L.); 2Engineering Research Center of Intelligent Computing for Complex Energy Systems, Ministry of Education, Baoding 071003, China

**Keywords:** virtual power plant, distributed energy, blockchain, consensus algorithm

## Abstract

Virtual power plant (VPP) composed of a large number of distributed energy resources (DERs) has become a regional multienergy aggregation model to realize the large-scale integration of renewable energy generation into the grid. Due to the characteristics of centralized management, the existing energy operation mode is difficult to simply apply to distributed energy resources transactions. The decentralization, transparency, contract execution automation and traceability of blockchain technology provide a new solution to the aggregation of decentralized resources and the opacity of transactions in VPP. In this paper, the existing problems of virtual power plants are analyzed, and the virtual power plant trading model is designed, which realizes the transparent benefit distribution and message transmission of virtual power plants. The virtual power plant blockchain network based on blockchain technology in this model solves the DERs coordination problem in VPP and the security and efficiency problems in information transmission. Combined with the actual situation of virtual power plant, the blockchain network collaboration mechanism (BNCM), which is convenient to reach agreement, is designed. Compared with the traditional practical Byzantine fault tolerance (PBFT) consensus algorithm, this mechanism can make DERs reach a consensus quickly. Finally, simulation experiments on the consensus algorithm show that the algorithm can reduce the collaboration time between DERs under the premise of ensuring the same fault tolerance rate and is more suitable for VPP scenarios with a large number of DERs.

## 1. Introduction

Virtual power plant (VPP) is used to realize the aggregation and coordination optimization of distributed generator, energy storage system, controllable load, electric vehicles and other distributed energy resources (DERs) through advanced information and communication technology, to participate in the power market and power grid operation as a special power plant [1]. VPP can effectively reduce fossil energy consumption, solve environmental problems such as carbon emissions, and provide system-level support services for energy trading [2]. 

VPP has no specific constraints on the geographical location and composition of DERs [3,4]. While using current technology and management methods, DERs will expose the distribution network to more uncertainties and increase its operational risks when DERs participates in VPP transaction [5,6]. Therefore, it is necessary to supervise DERs to a certain degree.

Researchers are eager to propose a highly flexible and adaptable DERs management method. Tomasz Sikorski et al. analyzed the technical and economic possibility of distributed energy integration into the virtual power plant and evaluated the economic efficiency of the virtual power plant model [7]. Aiming at the problem that the high penetration of distributed energy will bring great challenges to the existing centralized energy management methods, Yan et al. proposed a fully distributed energy management algorithm based on diffusion strategy [8]. Faced with the significant growth of interconnected distributed energy systems, Li et al. proposed an optimal coordination strategy among multiple distributed energy systems [9]. In [10], in order to manage intermittent distributed energy such as wind power, a novel VPP power supply strategy is proposed in this paper. In [11], fog as a virtual power plant is proposed to integrate power of distributed renewable power generators and the utility for a community. In [12], a novel structure of a power-to-gas-based virtual power plant is designed, which realizes the maximum utilization of clean energy and obtains high economic benefits. For the coordinated operation of distributed energy, Mahdi Rahimi et al. introduced virtual power plants as a solution and power plants to meet the needs of power and thermal load [13]. For the renewable energy with great potential of wind and solar energy, Giovanna Cavazzini et al. established a virtual power station model composed of pumped storage power station and wind farm [14]. However, in the above literature, DERs are uniformly dispatched by virtual power plant operators. With the increase in the number of DERs in VPP, the time cost, management cost, and transaction cost of VPP due to trust problems have increased dramatically [7]. Guided by the Internet thinking, the concept of regional distributed energy Internet has emerged in this context by constructing the interconnection network of multiple DERs in a certain region.

As one of the hottest concepts in the Internet, blockchain technology has the characteristics of being decentralized, transparent, and distributed, which can be used to solve the trust problem in virtual power plants. Blockchain is essentially the integration and innovation of distributed data storage, point-to-point transmission, consensus mechanism, encryption algorithm, and other computer technologies [15]. Brilliantov et al. believe that blockchain technology is one of the most effective ways to communicate in the future between virtual power plants and microgrids [16]. In [17], this paper discusses the feasibility of the application of blockchain in virtual power plant and designs a distributed optimization algorithm to manage VPP systems. Aiming at the problems of insufficient scalability, vulnerability to network attacks, and low processing efficiency in the traditional centralized energy trading mode, Guan et al. proposed a distributed energy trading scheme based on blockchain [18]. By analyzing the potential problems and challenges faced by peer-to-peer energy trading, Anker Lohachab et al. proposed a blockchain-based secure peer-to-peer energy trading framework for licensing smart cyber-physical systems [19]. In order to improve the intelligence, and security of direct transactions between Distributed energy generation company and users, Hu et al. proposed a blockchain-based IIoT peer-to-peer DE transaction model [20]. Bin et al. analyzed the existing ways and disadvantages of distributed energy transactions, studied the applicability of blockchain technology to distributed energy peer-to-peer transactions, and established a distributed energy settlement mechanism supporting distributed energy peer-to-peer transactions [21]. The above literature has deeply studied the application of blockchain technology in virtual power plants, but there are still some common problems.

The first problem is that the transaction is not transparent. VPP is used to realize the exchange of information and data on the power generation side, demand side, and power trading market, but it lacks a transparent benefit distribution mechanism. DERs and VPP operators cannot form a transparent choice of information, which increases the credit cost in the electricity transaction process. 

The second problem is the slow transaction speed. The use of blockchain in VPP has solved the security problem to a certain extent. However, the speed of node consensus in blockchain is slow. For example, the world’s largest blockchain network Bitcoin only supports 6–7 transactions per second, and it takes an hour to reach a final consensus for the transaction [22]. Moreover, the transaction storage in the blockchain has a certain delay, and the transaction information cannot be stored immediately [22]. Traditional blockchains are not suitable for VPP scenarios. 

Based on the above problems, we established a virtual power plant trading model (VPPTM) and designed the operation process of VPPTM. The virtual power plant blockchain network (VPPBN) based on blockchain technology in this model solves the security and efficiency problems in the information transmission of DERs in VPP. Finally, based on the actual situation of virtual power plants, this paper designs a blockchain network collaboration mechanism (BNCM). The mechanism consists of the fast consensus stage and the practical byzantine fault tolerance (PBFT) consensus stage. Compared with practical Byzantine fault tolerance (PBFT) consensus algorithms, this mechanism enables DERs to reach consensus quickly. The results of simulation experiments on the algorithm show that BNCM can achieve shorter cooperation time and better performance. This mechanism is more suitable for virtual power plant scenarios with many DERs.

The rest of this paper is organized as follows. Section 2 introduces a virtual power plant transaction model designed in this paper, and Section 3 introduces the blockchain network cooperation mechanism designed to facilitate agreement. Section 4 carries out relevant experiments and analyzes and Section 5 concludes the paper.

## 2. Virtual Power Plant Trading Model

VPP externally participates in the regulation of the power market as a whole, and internally realizes the coordinated operation control of each DERs. At present, the trust problem caused by the nontransparent information, the non-open rules, and the non-timely subsidies in the process of distributed energy trading has been paid more attention. For example, some energy service providers or load aggregators, after grasping the information on the national subsidy issuance policy in advance, forged information to defraud them [23]. In addition, users falsify their own transaction and electricity data to defraud high subsidies, and the transaction volume caused by power loss in electricity market transactions does not match the actual received volume, etc. The blockchain technology with the characteristics of decentralization, traceability, transparency, and tamper-proof can solve the trust problems mentioned above.

Therefore, in order to realize the transparency of information and benefit exchanges between DERs and VPP operators, this paper presents an improved virtual power plant trading model (VPPTM), through which the DERs in the virtual power plant construct the flow of transactions. Section 1 of this chapter introduces the overall structure and operation process of VPPTM, and Section 2 introduces the VPPBN that relies on blockchain technology to achieve DER collaboration in the VPPTM. 

### 2.1. Composition and Operation of the VPPTM

As shown in Figure 1, VPP operators in VPP assume external responsibilities and enjoy the benefits of response. The VPPBN based on blockchain replaces VPP operators to assume internal responsibilities, including ensuring the identity authentication of each subject in the VPP, ensuring the security of information exchange between subjects, and stimulating energy trading between subjects. There are multiple VPP operators in VPP to ensure the redundancy of the VPP infrastructure.

Figure 2 shows the operation process of VPPTM based on blockchain. The electricity consumption unit uploads its own electricity consumption data to the electricity trading market through the smart meter [24]. Through the auxiliary services of the trading market, the electricity consumption information is calculated to form a scientific electricity demand. VPP operators predict the power generation and load in the next period based on weather conditions, historical data, and the latest electricity demand data submitted by the trading market, and broadcast the forecast results to the VPPPN. According to the data provided by the VPP operators, the VPPPN proposes a plan for power purchase or generation of the day. Then, DERs complete the construction of several transaction plans through smart contracts based on blockchain technology. Finally, DERs store all transaction plans within a specified time in their own blockchain through the consensus mechanism (BNCM) newly proposed in this article. During the operating period, each DER disseminates the operating data in the VPPPN in real time, and finetunes its energy consumption plans and power generation plans based on the data in the VPPPN. VPPTM can realize adaptive scheduling and operation. After the end of this period, VPPTM sends the power generation data and loads data to the virtual power plant operator. The virtual power plant operator analyzes the data and improves the coordination parameters of the next day to improve the coordination efficiency of VPPTM. 

### 2.2. Blockchain Structure VPPBN in Virtual Power Plant

VPPTM relies on blockchain technology to achieve DER collaboration, and its blockchain structure is virtual power plant blockchain network (VPPBN). This section first introduces the composition of VPPPN, and then introduces its operation process. Blockchain has the characteristics of tamper-proof and traceability [3], all DERs in VPPTM can query the historical data stored in the VPPPN, and each subject has the same authority to query data in the VPPPN. As shown in Figure 3, each DER proposes its own energy demand or power generation plan for the next period based on historical information, and then signs the power generation plan through smart contracts. VPP operators collect all power generation plans within a specified time and package them into collaborative planning blocks.

The structure of the collaborative planning block is shown in Figure 4. The listener of the collaborative plan includes the hash value of the previous block, the timestamp when the collaborative plan packager was packaged, and the Merkle root of the collaborative plan block. All generation plan information recently occurred in the VPPPN is stored in the block body, which includes ID, power generation quantity, location, transaction price, etc.

When VPPBN is running, the VPP operator is responsible for packaging the transaction plan block. Each DER participates in the energy bidding on the VPPBN, and the smart contract is automatically executed when the smart contract trigger condition is reached. In VPPBN, both parties involved in the transaction will disseminate the transaction plan that has been reached. Finally, the DERs reach a consensus on the validity of this transaction plan through the blockchain network collaboration mechanism (BNCM) proposed in this paper. After a specified time, VPP operators will package all transaction plans in this period into transaction plan blocks and spread them on the VPPPN. Each DER verifies the legality of the transaction plan block containing all the transaction plans at the specified time. If it is legal, the block is stored in its own ledger, and if it is illegal, it is discarded.

## 3. BNCM: Blockchain Network Collaboration Mechanism in VPPBN

The traditional trading model using the PBFT algorithm has the disadvantages of wasting energy and low system throughput [25]. A bitcoin system, for example, produces a 1 MB block every 10 min [22]. The bitcoin system makes about seven transactions per second [22], which is not appropriate in a distributed energy trading system that requires a lot of trading. Moreover, the traditional solution to the problem of distributed energy trading often uses access mechanism. This mechanism improves the throughput rate by controlling the number of participating nodes and is still a centralized transaction mode to some extent. Nodes are small and vulnerable to external attack. This paper designs a blockchain network collaboration mechanism (BNCM) within VPP to facilitate agreement.

### 3.1. PBFT Consensus Algorithm Analysis

Consensus algorithms are used to reach a consensus among nodes in the blockchain [26]. The practical Byzantine fault tolerance (PBFT) consensus algorithm is one of the most widely used consensus algorithms, which aims to solve the Byzantine Generals Problem. The Byzantine Generals Problem talks about the problem of reaching a consensus in a scenario where a small number of nodes are allowed to falsify the transmitted messages [27].

The nodes involved in reaching a consensus in the PBFT algorithm are divided into primary node and backup nodes. The primary node is responsible for receiving client requests and sending them to other backup nodes. The backup node is responsible for verifying the correctness of the received message and sending corresponding reply messages to the primary node and other backup nodes. 

The number of invalid or malicious nodes tolerated by the PBFT algorithm is *f*. In order to ensure the normal operation of the entire system, the PBFT algorithm needs to have 2*f* + 1 normal nodes. Therefore, the total number of nodes in the system is 3*f* + 1. In other words, the PBFT algorithm can tolerate less than 1/3 invalid or malicious nodes. The consensus process of the PBFT consensus algorithm is divided into five stages: request, pre-preparation, preparation, submission, and response [28]. Figure 5 shows the execution process of the PBFT algorithm. The specific implementation process is shown below.

Request: The client (represented by “C” in Figure 5) sends a transaction request to the primary node (represented by “0” in Figure 5).Pre-prepare: After receiving the client’s request, the primary node sorts the client’s request. Then, the primary node broadcasts a pre-prepared message to the other backup nodes (represented by “1”, “2”, and “3” in Figure 5).Prepare: After each backup node receives the pre-prepared message from the primary node, it checks the validity of the message. After the verification is passed, the backup node sends a preparation message to other nodes including the primary node. As shown in Figure 5, “1”and “2” represent the backup node that can normally send the preparation message, and “3” represents the backup node that cannot be broadcast due to the node’s internal error.Commit: After the primary node and backup nodes receive the preparation message, they will check the validity of the message. When the node receives messages that have passed the verification, it sends a commit message to other nodes including the primary node. In Figure 5, “0”, “1”, and “2” represent the primary node and backup nodes that can send the commit message normally, and “3” represents the backup node that cannot be broadcast due to the node’s internal error.Reply: After receiving the commit message, the primary node and the backup node will check the validity of the message. When the backup node receives commit messages that have passed the verification, it sends a reply message to the client. After the client receives reply messages, the consensus is successful.

PBFT algorithm is often used to solve the consistency problem in distributed systems [29]. Compared with POW algorithm and POS algorithm, PBFT algorithm is more suitable for private chain and alliance chain scenarios because of its strong consistency and does not require a lot of computational support. IBM’s Hyperledger Fabric network is the world’s first commercial blockchain technology platform. Hyperledger uses PBFT as a consensus algorithm between nodes [27]. The PBFT algorithm requires all participating nodes to cross-validate messages, which leads to a significant decrease in the consensus efficiency of the PBFT algorithm with the increase of nodes, so the PBFT algorithm is not suitable for the application scenarios of many nodes. 

### 3.2. Definition of BNCM

The blockchain network collaboration mechanism (BNCM) in VPPBN is composed of the fast consensus stage and the practical byzantine fault tolerance (PBFT) consensus stage. The overall operation logic of the BNCM is shown in Figure 6. Each DER reaches a consensus through the fast consensus stage. Compared with the traditional PBFT algorithm, DERs can reach a consensus quickly in this stage. If the fast consensus stage is unsuccessful or times out, the PBFT consensus algorithm is used in the PBFT consensus stage to reach a consensus. 

The traditional PBFT consensus algorithm is usually used to solve the consistency problem in distributed systems [28]. The collaborative mechanism BNCM proposed in this paper is composed of two parts: the fast consensus stage and the PBFT consensus stage. The PBFT consensus stage is enabled only when the fast consensus stage is unsuccessful or timeout occurs. Compared with the traditional PBFT algorithm, the fast consensus stage of this paper has the following improvements. 

Cancel the pre-prepare step in PBFT algorithm

Because the cost of delegating DERs in PBFT to participate in virtual power plant is very high, the pre-prepare step in PBFT is cancelled in the fast consensus stage. If the DERs cannot reach consensus, then the PBFT consensus algorithm is adopted to reach the consensus through the PBFT consensus stage. 

2.VPP operators act as packers of transaction plan blocks

The main node in PBFT algorithm is selected by view number and number of nodes through fixed algorithm. If the selected master node is a malicious node, it will appear that the node does not broadcast the client request message or overtime broadcast, which will trigger the overtime mechanism of the PBFT algorithm and reselect the master node. If the reselected master node is still malicious node, the next round of reelection will continue, which greatly reduces the efficiency of the algorithm and is not conducive to the operation of virtual power plants [30]. In this paper, the VPP operator is used as the fixed selected master node in the fast consensus stage and the PBFT consensus stage of this algorithm, which is responsible for receiving the transaction plan in the virtual power plant and packing the transaction plan blocks, and then forwarding messages to all DER nodes.

3.In the rapid consensus stage, the virtual power plant operator verifies the response message reliability as the main node rather than the cross-validation of response message reliability by each DERs.

In the prepare stage and commit stage of PBFT algorithm, each replica node needs to synchronize point-to-point consensus with other nodes. However, when many distributed energy nodes in the virtual power plant communicate point-to-point, the communication time cost of broadcasting before nodes will increase exponentially [30]. Combined with the actual scene of VPP, the virtual power plant operator verifies the reliability of response message as the main node rather than the cross-validation of response message reliability by each DERs. When the virtual power plant operator cannot judge the authenticity of the message, the PBFT consensus is enabled. In most cases, DER collaboration time is reduced. 

### 3.3. Operation Process of BNCM

The blockchain network collaboration mechanism (BNCM) in VPPBN consists of the fast consensus stage and the PBFT consensus stage. Each DER first reaches a consensus through the fast consensus stage, and the flow chart of this stage is shown in Figure 7. If there are malicious nodes that make it impossible to successfully reach a consensus in the fast consensus stage, each DER then reaches a consensus through the practical Byzantine fault tolerance consensus stage. The flowchart of the PBFT consensus stage is shown in Figure 8.

Because the PBFT algorithm can tolerate less than *f* invalid or malicious nodes, the total number of nodes in this system is set to 3*f* + 1. The specific operation process of BNCM consists of eight steps, among which steps 1 to 4 are the fast consensus stage, and steps 5 to 8 are the PBFT consensus stage. 

Message preprocessing

The virtual power plant operator will summarize the transaction plan information collected from the VPPPN, and then package the information collected within the specified time interval into the transaction plan block. Finally, the virtual power plant operator broadcasts the block in the system.

2.DERs send response messages

After each DER receives the transaction plan block, it verifies the correctness of the block. If it is not correct, the block is discarded. If the verification is valid, each DER adds its own number to this block, and then sends a response message < *received*, *n*, *d*, *I* > to the virtual power plant operator.

Where *n* is used to sort the requests of the virtual power plant operator; *d* represents the summary of the transaction information in the transaction plan block, and *I* represents the current DER’s own number.

3.The virtual power plant operator counts the number of response messages

When 3f+1 response messages are received within the specified time interval, the virtual power plant operator believes that a consensus has been reached and sends an approval message *< agree*, *n*, *d*, *I >* to each DER. If the virtual power plant operator does not receive 3f+1 response messages within the specified time interval, indicating that the rapid consensus stage failed to reach a consensus, it skips to (5) and executes the PBFT consensus stage.

4.DERs process approval messages

After receiving the approval message, DERs update the transaction plan block to their own system. Consensus is completed.

5.Pre-prepared phase

When a consensus is not successfully reached in the fast consensus phase, the virtual power plant operator reaches it through the PBFT consensus stage. First, the virtual power plant operator broadcasts a pre-prepared message *< pre-prepared*, *n*, *d >* to each DERs, where *n* is used to sort the requests of the virtual power plant operator; *d* represents the summary of the transaction information in the transaction plan block.

6.Preparation phase

When each DER receives a pre-preparation message from the virtual power plant operator, each DER checks the correctness of the message. If it is not correct, the message is discarded. If the message is verified correctly, the DER sends a preparation message < *prepare*, *n*, *d*, *I* > to other nodes and virtual power plant operators, where *I* represents the current DER’s own number.

7.Commit phase

After the virtual power plant operator and other DERs receive the preparation message, they verify the correctness of the message. If it is not correct, the message is discarded. If 2*f* + 1 successful preparation messages are received, each DER sends a commit message < *commit*, *n*, *d*, *I* > to other nodes and virtual power plant operators.

8.Reply phase

After receiving the submitted message, virtual power plant operators and DERs check the correctness of the message. If it is not correct, the message is discarded. If DER receives 2f+1 verified commit messages, it indicates that most nodes in the current network have reached a consensus. The DER then updates the transaction plan block to its own system and returns < *reply*, *n*, *d*, *I* > to the virtual power plant operator. If the virtual power plant operator receives f+1 reply messages, the virtual power plant operator believes that this round of consensus has been reached.

It can be seen from the above specific operation process that in steps 1 to 4 of the fast consensus stage, the virtual power plant operator needs to receive the response messages of 3f+1 DERs before reaching a consensus. Since the existence of f malicious nodes are allowed, in the PBFT consensus stage of steps 5 to 8, the virtual power plant operator and each DER need to receive 2f+1 corresponding response messages in the preparation and submission stages before proceeding to the next step.

## 4. Experiment and Analysis

### 4.1. Design Realization of VPPTM Based on Ethereum

In this chapter, we use Ethereum to realize VPPTM. This chapter first introduces Ethereum technology, and then simulates the functions of power transaction and ledger maintenance in VPPTM through this technology.

Ethereum is one of the largest blockchain platforms in the world. It firstly proposed the smart contract concept [31]. The smart contract is a piece of code that triggers execution when both parties trade on the blockchain. In essence, the goal of Ethereum is to introduce the three major characteristics of blockchain technology: decentralization, openness, and security into almost all fields that can be calculated. Ethereum is divided into two types which are called public blockchain Ethereum (PUBBE) and private blockchain Ethereum (PRIBE). Anyone can join the PUBBE, while only permitted companies can join the PRIBE [31].

In this paper, we use Go-Ethereum client to implement the bottom layer of VPPTM, using Ethereum-browser to implement the data visualization in VPPTM. When finishing the building of programming environment, we set the parameter value of genesis block as Table 1. The genesis block refers to the first block in the blockchain, and it is generally used for initialization. As shown in Table 1, the gas limit value is set to 0×2fefd8, and the time stamp is set to 0×00. At the same time, we set the difficulty of mining blocks to 0×20000, which is much lower than the difficulty of PUBBE. Therefore, it is easier to generate new blocks and facilitate experimentation.

The pseudocode of this smart contract is shown in Table 2. There are three roles in the process of power transaction: power supplier, power user, and message maintainer. In the smart contracts of this section, power suppliers and power users refer to DERs who participate in completing a transaction, and message maintainers refer to VPP operators. When a smart contract is deployed to the Ethereum blockchain, it first needs to be compiled into machine language, and then the address of the smart contract is broadcast between VPPTM. When the smart contract is built, it will automatically call the instantiation functions of the three roles to realize the encapsulation and automatic execution of the block. The triggering condition of the smart contract is reached when the power user, the power supplier, and the message maintainer call their respective contract functions. At this time, the smart contract will be executed automatically. We use Ethereum Studio as programming tool in order to make the program more convenient, and the interface of programming is shown in Figure 9.

After VPPTM has been running some time, blockchain stores all electricity energy trading data and maintenance data, which are shown in Table 3. This table shows that the highest height of this block is 159, and every block has its hash value. “height of the block” refers to the block number. In other words, it is the number of blocks between a block and the genesis block. The hash value of the blockchain refers to the information that can uniquely and accurately identify a block. Any node in the blockchain can obtain the hash value of this block through a simple hash calculation. The difficulty of mining in this paper is much less than the mining difficulty of public chain Ethereum, so VPPTM creates blocks every second. Since counts of transactions in every second are not the same, the sizes of blocks generated are also always changing. Message maintainer can obtain some rewards when a block is successfully created; this mechanism also improves the fairness and security of VPPTM.

### 4.2. Design Realization of BNCM Based on Ethereum 

In order to verify the feasibility and performance of BNCM, an experimental simulation system is designed. We used Docker to simulate multiple DERs interactions in a single machine. Docker is an opensource application container engine based on Go language, which can easily package simulated DERs into transplantable container, and achieve complete virtualization by sandbox mechanism. Each Docker instance simulates a DER with the Docker version number of Docker enterprise 3.0. The experimental system is shown in Figure 10.

The physical server uses i7 7700k processor and 32 GB of running memory. The server operating system uses Ubuntu 16. 04 LTS. The experiment calls TcpDump to collect and analyze port network data, and calls ODBC library to write the collected data to MySQL database. In BNCM, Docker thread is used to simulate DERs and VPP operators. The DER port listens for the message, and after receiving the message, it passes the message to the core logic layer to judge the validity of the message and make corresponding processing. All historical data of DERs are stored in the account book. The intelligent contract is used to connect the DER interface layer, logic layer, and data layer. When the contract execution conditions are reached, the connect will be executed automatically. 

As shown in Figure 11, multiple containers are started through Docker to simulate the communication between DERs. In this experiment, the format of messages sent between DERs is specified as < status, data, send, received >. In the content of the message, status represents the status of the current message, including pre-prepared, prepare, commit, reply, etc. Data refers to the specific content of the message. Send refers to the sender of the message and received refers to the receiver of the message. After reaching a consensus, each DER stores the consensus result in its own blockchain. The consensus process is over. In this experiment, we assume that one virtual power plant operator and four DERs are included in the VPPTM. We tested the feasibility of reaching consensus in VPPTM by modifying the status of virtual power plant operators and DERs. By increasing the number of DERs and other behaviors, we then tested the consensus time between DERs in VPPTM, the fault tolerance performance of DERs, and other indicators.

By using the experimental environment, this section tests the BNCM coordination time, fault tolerance, and other indicators, and compares them with the PBFT algorithm.

#### 4.2.1. Collaboration Time Comparison Experiment 

In order to verify the efficiency of the consensus mechanism, a total of one VPP operator (A^OPE^) and four DERs (B^DER^, C^DER^, D^DER^, E^DER^) are set up in the experiment, and the communication link between each DER and VPP operator is normal. After running the BNCM and PBFT algorithms sequentially for 50 times, the time it takes to reach a consensus is counted in Figure 12.

As shown in Figure 12, BNCM can reach a consensus faster than the PBFT algorithm in the same experimental environment. After taking the average of the experimental data, it is found that the average time for BNCM to reach a consensus is 2.5992 s, while the average time for the PBFT algorithm is 5.2409 s. Therefore, through the experiment, the BNCM coordination mechanism can reach a consensus faster and more efficiently than the PBFT algorithm.

#### 4.2.2. Fault Tolerance Experiment

Fault tolerance refers to the maximum number of faulty nodes that the system can tolerate. It is related to the total number of nodes in the system. The more the total number of nodes, the more fault nodes can be tolerated. If the number of fault nodes exceeds the tolerable limit in the system, consensus will not be reached.

In order to test the fault tolerance of BNCM in VPPTM, a virtual power plant operator and nine DERs were set up in this experiment. In other words, the number of nodes participating in reaching a consensus in the system is 10. Let the number of fault nodes be set to 0, 1, 2, 3, 4, and 5 in turn, and count the time it takes to reach a consensus.

It can be seen from Figure 13 that when the number of fault nodes is 0, 1, 2, 3, the system can reach a consensus in a limited time, but when the number of fault nodes is greater than or equal to 4, the consensus cannot be reached smoothly. Through this experiment, it is found that the fault tolerance performance of BNCM is the same as that of PBFT. In other words, when there are 3*f* + 1 nodes in the system, at most, *f* fault nodes are allowed.

#### 4.2.3. Comparison Experiment of Cooperation Time in the Presence of Communication Failure Nodes

To verify the efficiency of BNCM when no consensus is reached in the fast consensus stage, this section conducts the experiment based on Section 4.2.1. A total of one VPP operator (A^OPE^) and four DERs (B^DER^, C^DER^, D^DER^, E^DER^) are set up in this experiment. Because fault tolerance experiments show that when there are 3*f* + 1 nodes in the system, at most, *f* malicious nodes are allowed. Therefore, this experiment conducts a comparative experiment in the case of one malicious node in the system.

In this experiment, the communication link of B^DER^ is set to have 20%, 40%, and 60% probability of failure by modifying the state of B^DER^. When a communication failure occurs, B^DER^ will not be able to communicate with other nodes normally. Experimental results when B^DER^ has 20%, 40%, and 60% failure rates are shown in Figure 14a–c.

The experimental results in Figure 14a–c show that the time for the PBFT algorithm to reach a consensus shows an overall stable trend. However, the BNCM shows a fluctuating trend. After analyzing the reasons, it is found that when B^DER^ can communicate normally, BNCM reaches a consensus through the fast consensus stage. Therefore, the consensus time is shorter than the traditional PBFT algorithm. When a communication failure occurs in B^DER^, the fast consensus phase of BNCM fails, and each DER reaches a consensus through the PBFT phase. Therefore, the time to reach a consensus is longer than the traditional PBFT algorithm.

Finally, after analyzing the experimental data, it is found that the average time for BNCM to reach consensus is 3.598 s, 4.615 s, and 5.579 s in Figure 14a–c. The time for the PBFT algorithm to reach consensus has been stable at 5.261 s. We can infer that when the failure rate of B^DER^ increases, the advantage of BNCM slowly diminishes. When the failure rate of B^DER^ reaches 60%, the PBFT algorithm is more efficient. We can conclude that when the failure rate of *f* nodes in the system increases, the efficiency of BNCM will gradually decrease. Eventually, BNCM will lose its efficiency. Overall, BNCM is more efficient than traditional PBFT algorithms. 

#### 4.2.4. Comparative Experiment on the Increase of the Number of DERs

This section tests the relationship between the collaborative completion time and the number of DERs under different mechanisms. By recording the average value of the consensus process of 30 times, the PBFT algorithm and the BNCM are compared when the number of consensus nodes increases from one to eight.

As shown in Figure 15, the consensus completion time of the PBFT algorithm is longer, while the consensus completion time of BNCM is shorter. As the number of DERs increases, the advantages of BNCM become more obvious. When the number of nodes in the experiment is eight, the average consensus completion time of the PBFT algorithm is 9.2 s, and the average completion time of BNCM is 2.99 s. Through the experiment, it can be concluded that the performance of BNCM is better than that of PBFT algorithm.

#### 4.2.5. Analysis of Experimental Results

In the collaborative time comparison experiment, the collaborative speed of DERs in BNCM is significantly shorter than that of PBFT. In the actual scene of VPP, due to the long distance between DERs and the higher error rate of data transmission than the single machine environment, the synergy effect of BNCM will be more advantageous in the actual situation. The fault tolerance experiment shows that when there are three DERs in BNCM, malicious DERs can be allowed at most. The fault tolerance performance of BNCM is the same as that of PBFT. Comparative experiments on the scale of DERs show that with the increase in the number of DERs, the synergy performance of BNCM is higher than that of PBFT.

## 5. Conclusions

The innovation of this paper mainly has the following aspects. 

Through the analysis of the problems existing in the virtual power plant, the virtual power plant trading model is designed, which realizes the transparent distribution of benefits and message transmission in the virtual power plant.Combined with the advantages of blockchain technology, such as decentralization, transparency, contract execution automation, and traceability, this paper designs a virtual power plant blockchain network named VPPBN based on blockchain technology in VPPTM model, which solves the problems of DERs coordination, security, and efficiency in information transmission in VPP.Combined with the actual situation of virtual power plants, this paper designs a convenient and agreed internal DERs coordination mechanism in VPP named blockchain network collaboration mechanism (BNCM). The experimental results show that, compared with the current common PBFT, the collaborative mechanism can achieve shorter collaborative time and better performance.

The data in this paper are only obtained in the simulation experiment platform. In the future, we will continue to improve VPPTM and test this model in a real environment. The emergence of new technologies such as side chain [32] and multichain [33,34] may improve the performance of VPPTM, which needs further research in the future.

## Figures and Tables

**Figure 1 sensors-22-01783-f001:**
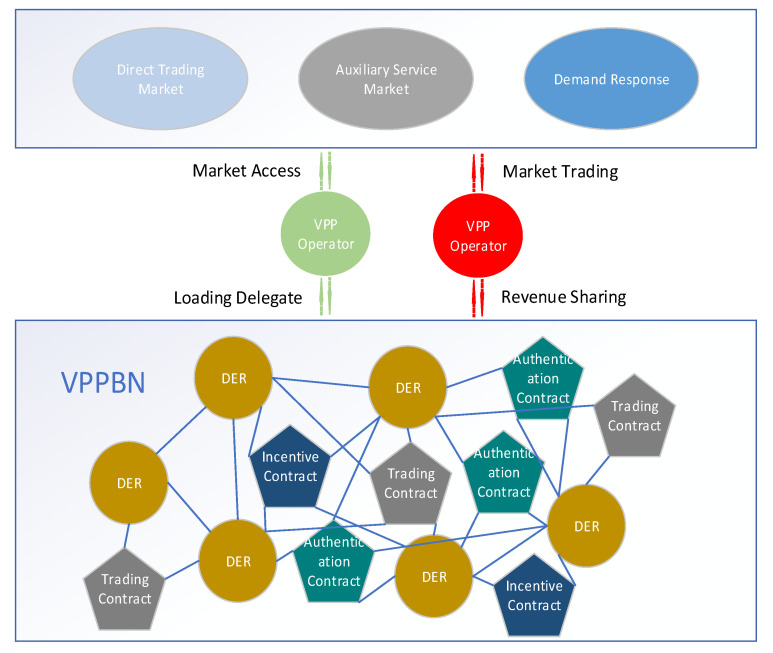
Virtual power plant trading model.

**Figure 2 sensors-22-01783-f002:**
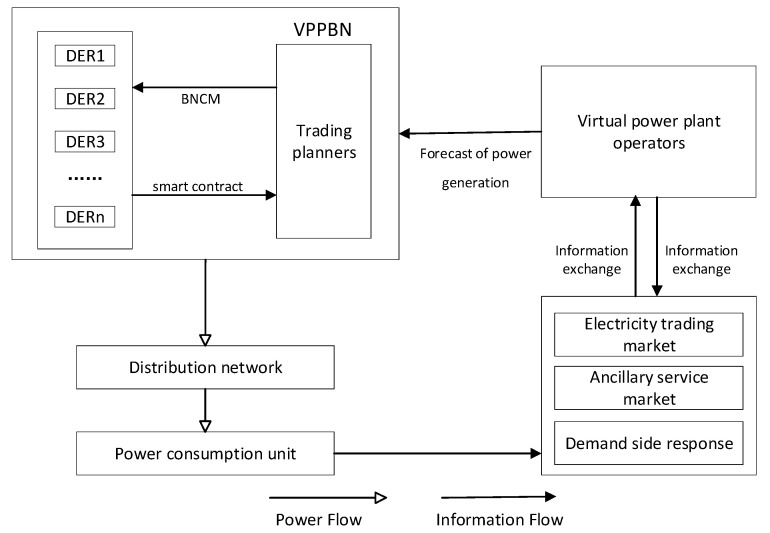
Working process of VPPTM based on blockchain.

**Figure 3 sensors-22-01783-f003:**
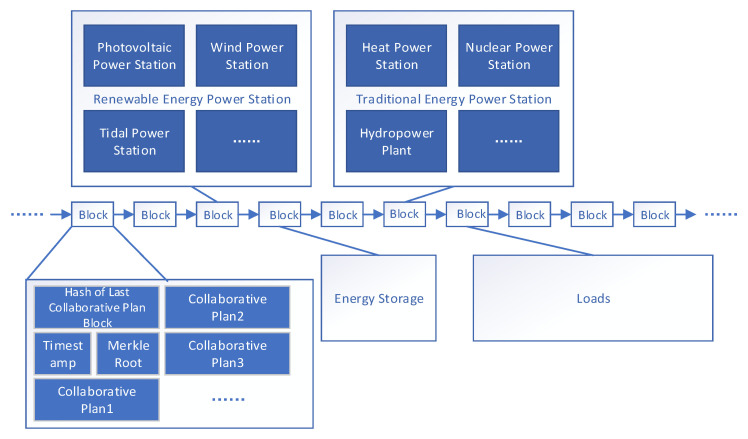
VPPBN structure.

**Figure 4 sensors-22-01783-f004:**
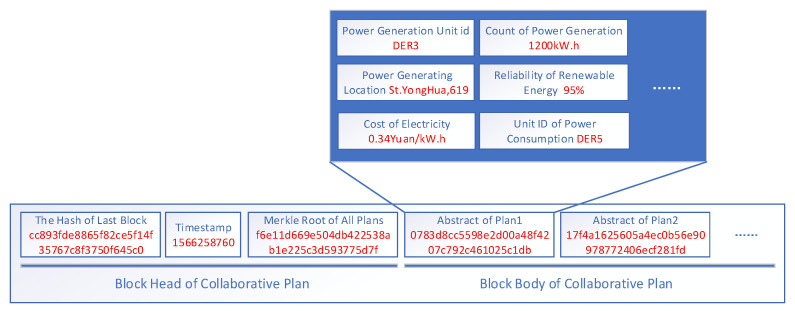
Structure of collaborative plan block in VPPBN.

**Figure 5 sensors-22-01783-f005:**
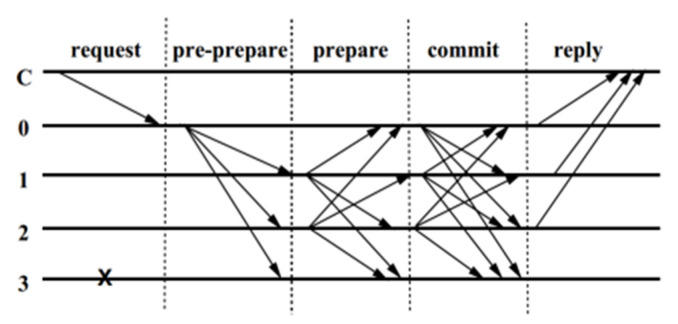
Execution process of the PBFT algorithm.

**Figure 6 sensors-22-01783-f006:**
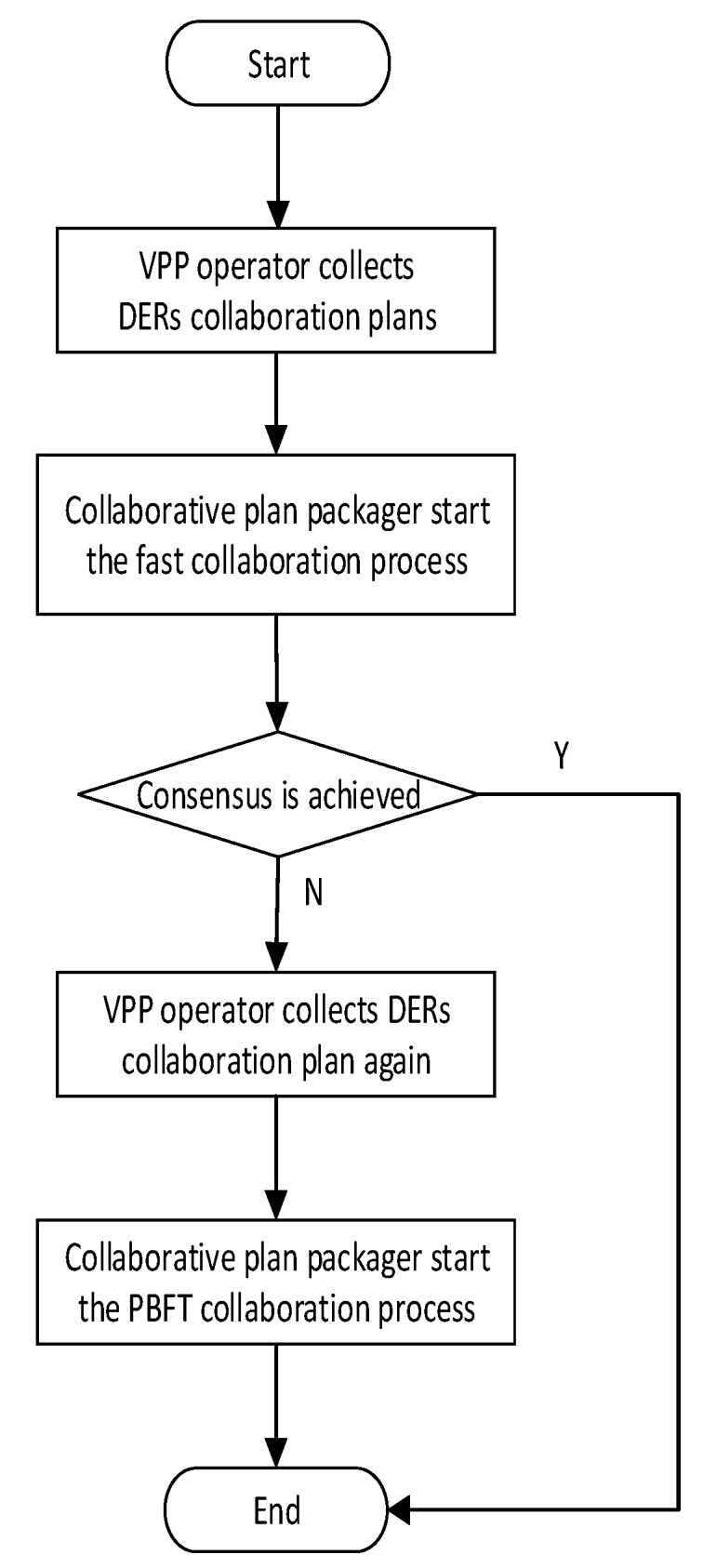
Operation logic of BNCM.

**Figure 7 sensors-22-01783-f007:**
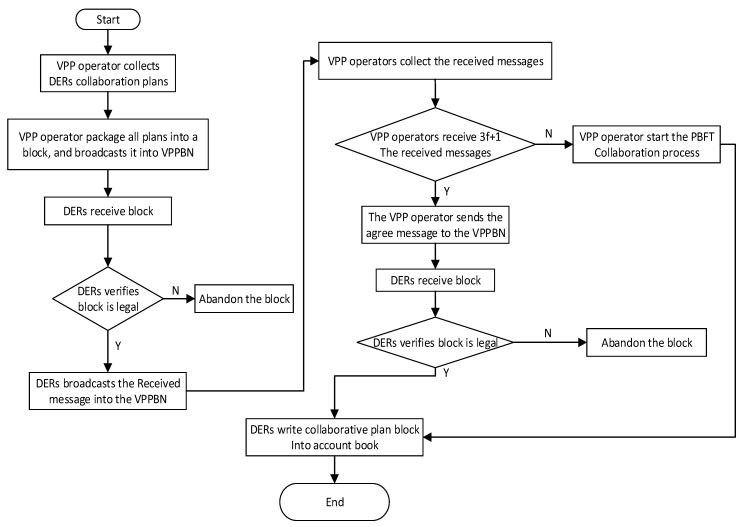
The flow chart of the fast consensus stage.

**Figure 8 sensors-22-01783-f008:**
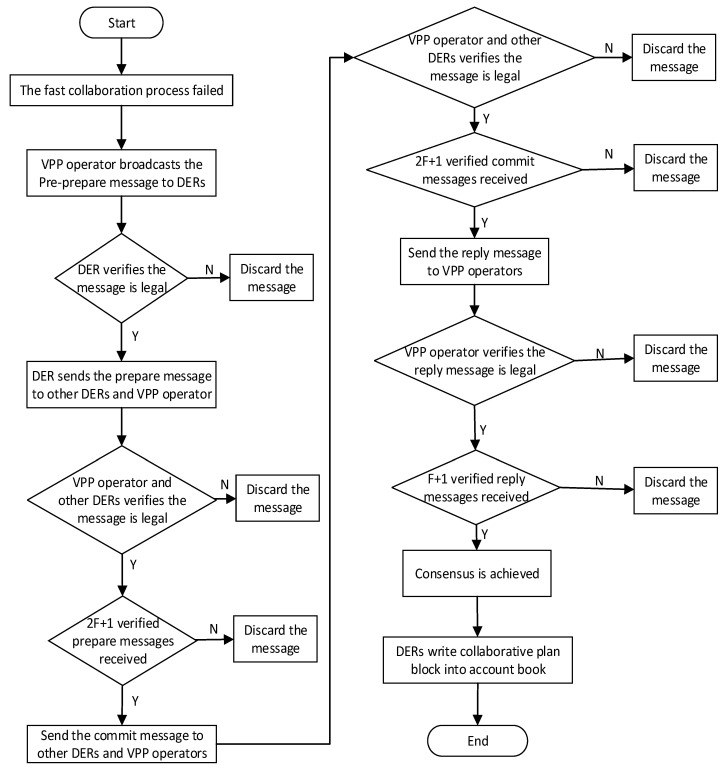
The flow chart of the PBFT consensus stage.

**Figure 9 sensors-22-01783-f009:**
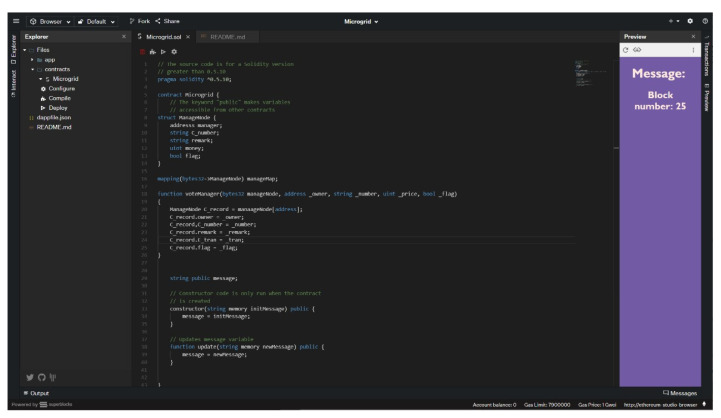
Experimental interface based on Ethereum technology.

**Figure 10 sensors-22-01783-f010:**
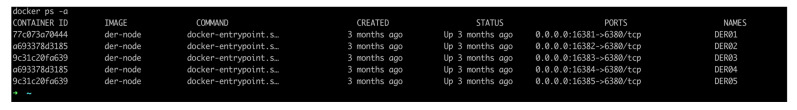
The deployment of projects based on docker.

**Figure 11 sensors-22-01783-f011:**
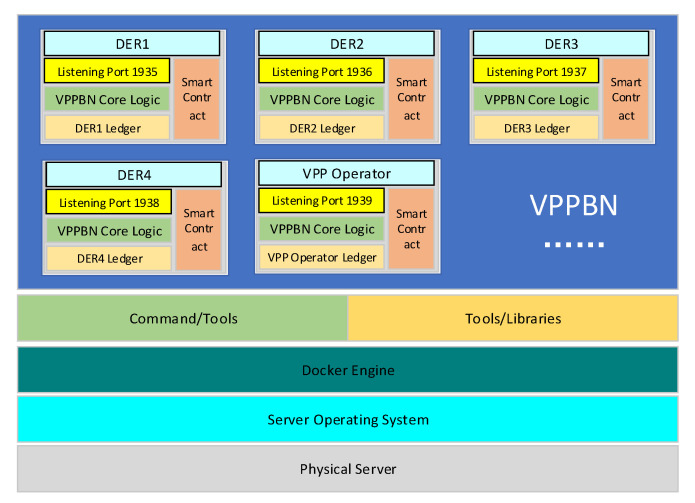
Experimental simulation system.

**Figure 12 sensors-22-01783-f012:**
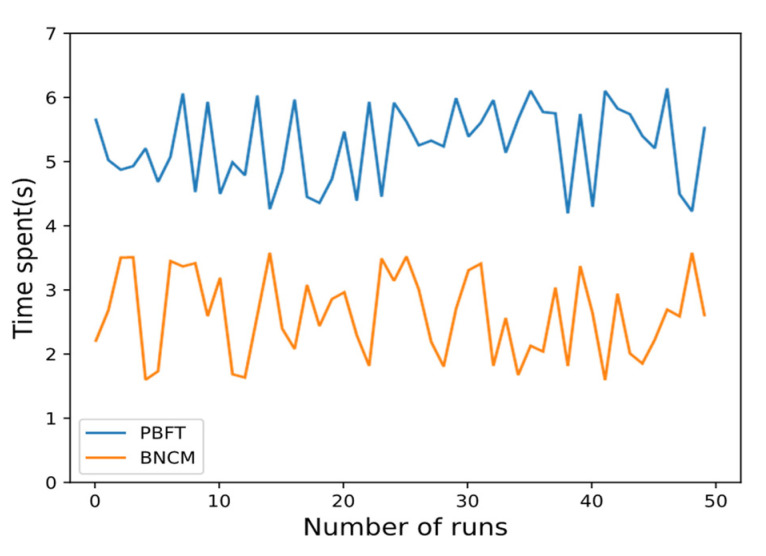
Comparison of VPPTM and PBFT collaborative processes.

**Figure 13 sensors-22-01783-f013:**
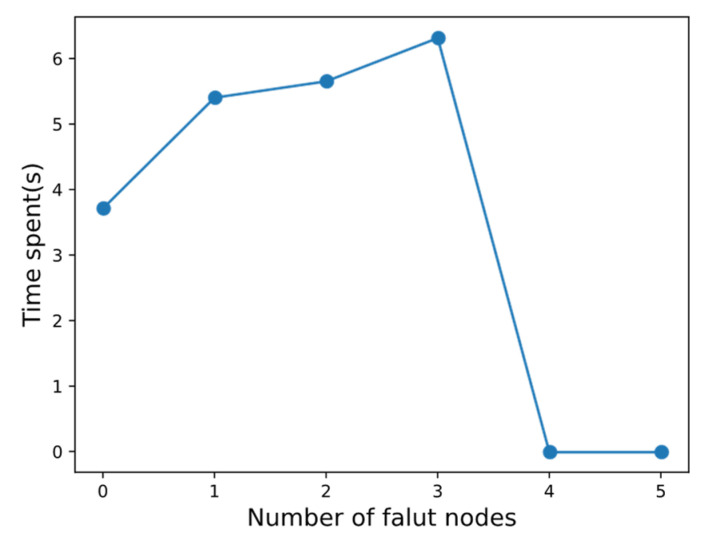
Fault tolerance of VPPBN.

**Figure 14 sensors-22-01783-f014:**
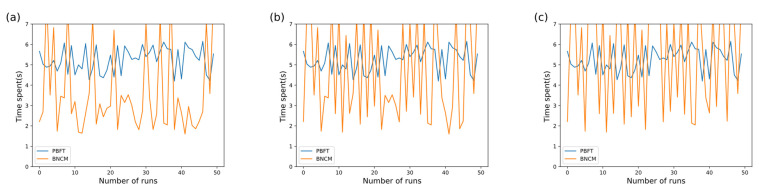
Comparison of VPPTM and PBFT collaborative processes. (**a**–**c**) B^DER^ has 20%, 40%, and 60% failure rates.

**Figure 15 sensors-22-01783-f015:**
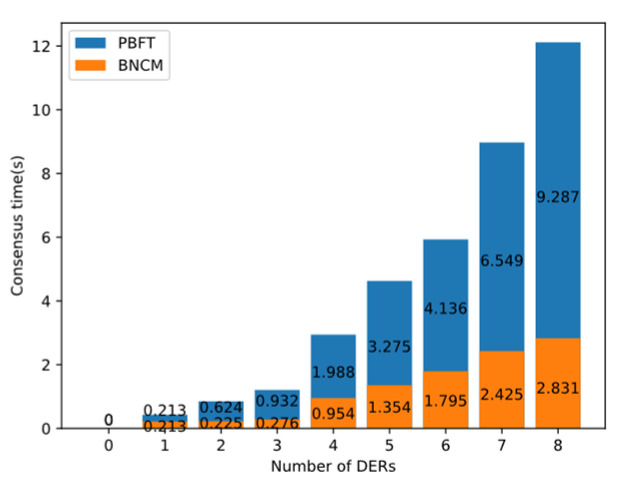
The relationship between DERs number and collaboration time under different collaboration mechanisms.

**Table 1 sensors-22-01783-t001:** Parameters and descriptions of genesis block.

Parameter	Value	Description
Chain ID	10	ID of blockchain view
Alloc	{}	Used to set login account
Coinbase	0×00000000000000000	Initial ledger
Difficulty	0×20000	Mining difficulty
Gas limit	0×2fed21	Gas limit of every transaction
Nonce	0×000000000000000042	64-bit random number used only once
Mix hash	0×000000000000000000	Hash value of current block
Parent hash	0×00000000000000000	Hash value of parent block
Timestamp	0×00	The timestamp of the genesis block generation

**Table 2 sensors-22-01783-t002:** Pseudocode of electricity energy trading contract.

Description
Commit request to the blockchain
Commit classification, reputation, address, and other parameters to the blockchain
Call function commit Response ()
Assign address of message Maintainer;
Assign name of message Maintainer;
Assign name of Power Supplier;
Assign name of Power User;
Assign remark of subjects like default possibility, trustworthiness;
Assign state of this trading process;
Deal reached
Increase the height of blockchain
Transaction data is written to the blockchain
Link final block to the blockchain

**Table 3 sensors-22-01783-t003:** Blockchain information of BNCM.

Height of Block	Hash Value	Generating Time	Miner	Block Size
159	14548fefadd6db899a1503ed9	3 min	Maintain1	2532 bytes
158	75e417d8425cbabd9a887fd8e	3 min	Maintain2	3785 bytes
157	ff2139c672957245740d1867b	3 min	Maintain1	1057 bytes
156	e9aaddcb34f25abe620b753fd	3 min	Maintain3	3738 bytes
155	4db44e074e4437d912b902c9	3 min	Maintain1	4201 bytes
154	1238ac1299ad1d7a7dda5000a	3 min	Maintain1	5053 bytes
153	3995ca97f3bd3e45ec09b841b	3 min	Maintain2	2464 bytes
152	21d4d243d922d18aae40e7b2	3 min	Maintain2	5437 bytes
151	14851ba6f85ac3d47622a45ca	3 min	Maintain3	4756 bytes
150	c9c81d4298f51281bfc7a70d4	3 min	Maintain2	8573 bytes
149	153f7bcb1263b70a20a636575	3 min	Maintain3	1587 bytes

## Data Availability

Not applicable.

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
