# Peer review of "Research on Distributed Energy Consensus Mechanism Based on Blockchain in Virtual Power Plant"

_sensors, 2022, doi:10.3390/s22051783_

Round 1
Reviewer 1 Report
The article presents a technique of "blockchain network collaboration mechanism (BNCM)" in order to improve the functioning of Virtual power plant (VPP). The article is well written with theoretical discussions and interesting quantitative results. I forward some specific comments in order to improve the presentation of the article:
- The authors quote in line 97: "Electricity cannot be stored". However, there are several energy storage systems.
- In the introduction, the authors explain the importance of VPP citing the works developed in recent years [lines 42-65]. Furthermore, the authors clearly indicate that blockchain technology can be used in VPPs, again citing several works in the literature [libes 66 - 87]. Finally, the authors highlight the problems/limits of current blockchain technology for application in VPPs [lines 88 - 98]. However, the initial discussion of the article's proposal/contribution is not cited in the introduction. The authors comment on lines 105 - 107 that this topic will be discussed in section 3. However, in order to present a "complete" introduction (with the problem and solution), I suggest that the authors insert a preliminary discussion/explanation about the proposal/contribution of the article still in the introductory section of the article.
- Authors use different formatting in the text, for example "Section 1" [line 114] and "Section II" [line 115].
- Figure captions are wrong (there are two Figures 1)
- Line 142, in Figure 2 (in the VPPBN box) the authors show four "DER1". Wouldn't it be DER1, DER2m DER3... DERn?
- The authors present in Section 2 the VPP model and blockchain technology applied to VPPs. However, no details of the problem cited in lines 88 - 98 were presented by the authors. As the article contribution is exactly on this topic, I suggest that the authors comment more on the problem to be studied. In other words, the problems must be properly discussed in the article, especially the problem of the slow transaction speed.
- The text on lines 183 - 186 is confused. Please rewrite that paragraph.
- The authors cite in line 231: "Compared with the traditional PBFT algorithm, DERs can reach a consensus quickly in this stage. If the fast consensus stage is unsuccessful or time out, the PBFT consensus algorithm is used in the PBFT consensus stage to reach a consensus." Some doubts arose, first how fast is the BNCM compared to PBFT? and how exactly can this agility help VPPs? Also, if BNCM can present consensus problems, requiring the use of PBFT in "background", wouldn't that be a limitation? How to guarantee agility in the process if there may be slower processes within the methodology? In other words, how to guarantee that the process is fast if there is no guarantee of consensus in the BNCM process?
- Figure 12 presents the objective result proving that BNCM is faster than PBFT. In fact this was expected since the BNCM flowchart (Fig7) is smaller than the PBFT flowchart (Fig8). However, the response time when BNCM does not reach consensus is not shown in Figure 12. I understand that Figure 13 does not show the comparison between the times of the two strategies. In this reviewer opinion, the authors need to prove that the response time of the proposed strategy is smaller (or minimally equal) to the response time of the conventional strategy. This is my biggest doubt, since Figure 6 indicates that the time can be longer in the case of non-consensus.
- In Figure 14 the authors present the result as VPPCM. I suggest change to BNCM.
Author Response
请参阅附件。

Reviewer 2 Report
This paper presents the application of Distributed Energy Consensus Mechanism Based on Blockchain to coordinate the operation of DERs located in VPP and to solve problems related to the security and efficiency in information transmission. Chapter 1 provides a detailed literature review. Chapter 2 presents aspects of the Virtual Power Plant Trading Model. Chapter 3 covers the Blockchain Network Collaboration Mechanism in VPPBN. Chapter 4 presents the experiment and analyses the results. The article ends with Chapter 5, which presents the conclusions.
The solution and the model presented in the paper is discussed in depth. The literature review is very detailed. The methodology is carried out correctly. The format of the article is also correct.
The following are general and specific comments.
General comments:
1) Contributions are not clearly stated. Preferably it should be at the end of the introduction, before the structure of paper.
2) Although the model is discussed in great detail, I can't see how the electrical problems were solved or dealt with. How the authors ensured proper voltage level, frequency level, which are the result of the power flow?
3) The article must be proofread for language before publication. The vocabulary used, sentence structure are often incorrect. This affects the way the article is received. It is difficult to read, one often has to read a paragraph several times to understand the sense of the written sentence.
Specific comments:
1) Lines 93-98: Reference is needed. Especially regarding the: "For example, the world's largest blockchain network Bitcoin only sup-95 ports 6-7 transactions per second, and it takes an hour to reach a final consensus for the 96 transaction." What is the source of this information?
2) Figure between lines 142 and 143 should be named Figure 2. That will also change the numeration of figures in the whole paper.
3) Fig. 9 - should be Figure. Also why the picture is mirrored?
4) Fig. 11 - should be Figure and without a bold
5) Lines 420-421: "Figure 3 counts the time it takes to reach a consensus each 420 time, in the process of using the BNCM and PBFT algorithms to run 50 times." - something is wrong with the sentence. Now it seems that the figure 3 is doing something. Try: "algorith shown on figure 3 ... "
Round 2
Reviewer 1 Report
All my comments were addressed by the authors.
I emphasize that the additional results (lines 462-487) show, more clearly, that the authors' proposal (BNCM) has an advantage over the conventional method (PBFT) up to a limit of failure rates (in this case of 60% according to the authors). This information does not detract the authors' contribution, on the contrary, it clearly indicates where the proposed method (BNCM) can be used in order to present a better efficiency.
Congratulations for the excellent work!
Reviewer 2 Report
The authors have addressed my comments.